# Distribution of Iron Nanoparticles in Arrays of Vertically Aligned Carbon Nanotubes Grown by Chemical Vapor Deposition

**DOI:** 10.3390/ma15196639

**Published:** 2022-09-24

**Authors:** Alexander V. Okotrub, Dmitriy V. Gorodetskiy, Artem V. Gusel’nikov, Anastasiya M. Kondranova, Lyubov G. Bulusheva, Mariya Korabovska, Raimonds Meija, Donats Erts

**Affiliations:** 1Nano Ray-T, Institūta Iela 1, Ulbroka, LV-2130 Stopiņu Novads, Latvia; 2Nikolaev Institute of Inorganic Chemistry SB RAS, 630090 Novosibirsk, Russia; 3Institute of Chemical Physics, University of Latvia, Jelgavas Iela 1, LV-1004 Riga, Latvia

**Keywords:** vertically aligned carbon nanotube arrays, catalytic chemical vapor deposition, iron nanoparticles, EDX analysis

## Abstract

Arrays of aligned carbon nanotubes (CNTs) are anisotropic nanomaterials possessing a high length-to-diameter aspect ratio, channels passing through the array, and mechanical strength along with flexibility. The arrays are produced in one step using aerosol-assisted catalytic chemical vapor deposition (CCVD), where a mixture of carbon and metal sources is fed into the hot zone of the reactor. Metal nanoparticles catalyze the growth of CNTs and, during synthesis, are partially captured into the internal cavity of CNTs. In this work, we considered various stages of multi-walled CNT (MWCNT) growth on silicon substrates from a ferrocene–toluene mixture and estimated the amount of iron in the array. The study showed that although the mixture of precursors supplies evenly to the reactor, the iron content in the upper part of the array is lower and increases toward the substrate. The size of carbon-encapsulated iron-based nanoparticles is 20–30 nm, and, according to X-ray diffraction data, most of them are iron carbide Fe_3_C. The reasons for the gradient distribution of iron nanoparticles in MWCNT arrays were considered, and the possibilities of controlling their distribution were evaluated.

## 1. Introduction

Carbon nanotubes (CNTs) are attractive as fillers that improve the mechanical and electrical properties of composite materials [1,2]. This led to the development of large-scale methods for the synthesis of CNTs. The market mainly includes entangled single-walled CNTs (SWCNTs) or multi-walled CNTs (MWCNTs) produced by catalytic chemical vapor deposition (CCVD). At the same time, the technology of CCVD synthesis of the arrays of ordered CNTs is of great interest. Materials based on CNTs aligned vertically on a substrate can be used in high-tech applications. They have been tested as cold cathodes for flat displays [3,4,5] and X-ray tubes [6,7,8,9], tip probes in scanning probe microscopy [10,11], electrode materials in supercapacitors and lithium-ion batteries [12,13,14], elements of biochemical sensors [15,16], and THz absorbers [17]. Recent reviews summarize the progress in the synthesis of vertically aligned CNTs [18] and their use in various fields [19], including biomedical applications [20]. 

The length, diameter, and defects of CNTs, their order and density in the array, and the presence of a pyrolytic carbon by-product and a metal, used as a catalyst in the synthesis process, affect the properties of the material. The structural characteristics of the array depend on the synthesis conditions, in particular, the composition and supply rate of the feeding gas, catalyst, substrate temperature, pressure in the reactor, etc. To reveal these dependencies, detailed studies were carried out [21,22]. The synthesis process is based on the decomposition of carbon-containing gases on catalytic particles. The most efficient and frequently used catalysts are carbide-forming iron group metals and their alloys. To obtain CNTs oriented vertically with respect to the substrate, the density of nuclei must be high enough to ensure mutual repulsion of growing CNTs [23]. In addition, good adhesion of the catalyst to the substrate is required for the root growth of CNTs.

Essentially, two methods are used to prepare a catalyst for the synthesis of CNT arrays. The first one is based on the preliminary deposition of a thin metal film on the substrate. The film thickness affects the size of metal catalyst nanoparticles [24]. In this case, the formation of nanoparticles with a narrow size distribution is possible, which ensures the growth of CNTs with similar outer diameters [25]. The dependencies of the diameter and number of layers in CNTs on the size of catalyst nanoparticles were studied in [26]. During synthesis, the metal catalyst is in a quasi-liquid state, and this provides thermodynamic and kinetic conditions for the formation of graphene layers [27]. The thickness of arrays formed on substrates with a pre-deposited metal film is limited by the catalyst consumption during CNT growth. One of the reasons for the loss of the catalyst is the capillary effect of liquid metal being sucked into the internal cavity of the CNT [28]. Typically, the thickness of the array produced in this way rarely exceeds 100 µm [29].

Another approach to the synthesis of arrays of vertically aligned CNTs is aerosol-assisted CCVD, where the feedstock is a solution of an organometallic compound in an organic solvent [30]. More often, solutions of ferrocene in hydrocarbons are used for this. When the aerosol reaches the hot substrate, the ferrocene decomposes, and the released iron atoms form nanoparticles. Hydrocarbon decomposition products penetrate into iron nanoparticles, and when iron is saturated with carbon, CNT growth begins [31,32]. Being in a liquid state during synthesis, the metal penetrates into the internal cavity of the CNT, and after cooling, the sample forms nanoparticles. These nanoparticles are mainly in the form of Fe_3_C and α-Fe [33] and impart magnetic properties to arrays [34,35,36,37]. Such arrays can be used as electromagnetic shielding and broadband absorber coatings, memory elements, magnetic markers [38,39,40], and membranes [41,42].

Aerosol-assisted CCVD synthesis of CNTs is technologically simple since there is no stage of deposition of the initial metal film. Another advantage is the continuous supply of a metallic source on the substrate, which prolongs the life of the catalyst. As a result, arrays several millimeters thick can be produced [43]. However, in such synthesis, it is necessary to coordinate the main parameters, in particular, the composition of the gas phase and the synthesis temperature, in order to have arrays of the required length, diameter, and ordering of CNTs [44,45,46]. 

The purpose of this work was to determine the main stages of aerosol CCVD synthesis of CNTs and to estimate the size and distribution of metal nanoparticles in an array produced from a ferrocene–toluene mixture. Among other possible liquid hydrocarbons, toluene is one of the best candidates for the mass production of vertically aligned MWCNTs [47]. The ability to dissolve up to 13 wt.% ferrocene at room temperature makes toluene attractive for obtaining arrays with various iron contents. Obviously, the content of the metal and its location in the array will affect the properties of the material. In particular, it has been shown that encapsulated iron nanoparticles interact with each other, which leads to the anisotropy of the magnetic properties of oriented MWCNTs [48,49,50].

## 2. Materials and Methods

The scheme of the CVD setup is shown in Figure 1a; the details of its construction and synthesis technique are given elsewhere [51,52]. The stainless steel tubular reactor has 800 mm in length and 42 mm in diameter. The central part of the reactor is located in a furnace, which can provide a heating temperature of up to 1000 °C. The manipulator and inputs for the carrier gas and the reaction mixture are connected to the input block using a quick coupling. The window allows one to observe the growth of the array and register images. A quartz tube 30 mm in diameter was inserted into the reactor, into which a quartz boat with (001) silicon substrates for CNT synthesis was placed using a manipulator. From one to ten substrates with a size of 10 × 10 mm^2^ can be used in synthesis. The setup is equipped with a device for injecting liquid into the synthesis zone. Direct injection makes it possible to achieve a high accuracy of dosing of the precursor and create high concentrations of reagents in the place of CNT nucleation and growth. 

A solution of ferrocene (4 wt%) in toluene was used as the reaction mixture. The solution was injected into the reactor zone at a temperature of above 200 °C, which is much higher than the boiling point of toluene. The argon flow delivered vapors to the substrates. The main synthesis parameters were studied earlier [53]. The optimal conditions for our setup are a temperature in the synthesis zone of 800 °C and a feed rate of the ferrocene–toluene solution of 15 mL h^−1^.

The structure of CNT arrays was studied using the methods of scanning electron microscopy (SEM) on Hitachi S-3000N Japan and Dual-beam FIB/FEI Helios 450S microscopes, USA and transmission electron microscopy (TEM) on a JEOL 2010 microscopem Japan. Raman spectra were recorded on a Horiba LabRAM HR Evolution spectrometer, Japan using an argon laser excitation of 514 nm with a power of 1 mW. X-ray diffraction (XRD) analysis was carried out at room temperature on a Shimadzu XRD-7000 diffractometer, Japan (CuKα radiation, Ni filter on the reflected beam, and a scintillation detector with amplitude discrimination). The data were collected with a 0.1° step size in the 2Θ range between 20° and 85°. The electronic state of carbon was studied using X-ray photoelectron spectroscopy (XPS) and near-edge X-ray absorption fine structure (NEXAFS) spectroscopy at the Russian–German dipole beamline of the BESSY II electron storage ring operated by Helmholtz-Zentrum Berlin für Materialien und Energie, Germany, Germany. The distribution of iron nanoparticles in CNT arrays was determined on a Hitachi S-4800 SEM microscope, Japan using an energy-dispersive X-ray (EDX) spectroscopy with a Bruker XFlash 5010 detector, Germany. 

## 3. Results

### 3.1. Synthesis of CNT Arrays

To estimate the growth rate of the CNT array at all stages of synthesis, the side image of the substrate was recorded using a photo-registration system (Figure 1a). Using a camera located on the side of the reactor input, every 2 min from the beginning of the synthesis until the next 40 min and then with an interval of 10 min, images of the sample were taken. The photographs taken at the beginning and after 40 min of synthesis are included in Figure 1b and 1c, respectively. The photographs show a semicircular end of a quartz boat and a silicon substrate heated to a working temperature of 800 °C. Comparison of the change in the sample thickness with respect to the substrate thickness made it possible to plot the dependence of the CNT array thickness on the synthesis time (Figure 1d).

The obtained S-shaped dependence can be divided into three sections. The first 7 min stage corresponds to the formation of iron nanoparticles, their saturation with carbon, and the beginning of CNT synthesis. The second stage between 10 and 50 min is a uniform growth of the CNT array. In the next final stage, the CNT growth rate decreases. The catalytic layer is formed on the substrate surface in the first stage of synthesis. This process includes the agglomeration of iron into nanoparticles, the saturation of these nanoparticles with carbon, and the formation of iron carbides and silicides. The latter species affects the adhesion of catalyst nanoparticles to the substrate. Some of the nanoparticles are covered with graphitic carbon and become the bottom base of the growing array of CNTs. In the stage of intensive growth of the array, the catalytic layer absorbs carbon from the gas phase and transforms it into CNT layers. The iron coming from the decomposition of ferrocene feeds the catalytic layer, which loses iron due to its capture into the internal cavities of CNTs. An array with a thickness of ~500 μm shields the substrate surface from the penetration of hydrocarbon and ferrocene vapors. A decrease in the supply rate of reagents causes a decrease in CNT growth. Specific synthesis parameters and the CNT density on the substrate determine the conditions under which this decrease begins [54].

### 3.2. Characterization of CNT Arrays

SEM images of the side and surface of a CNT array grown on a silicon substrate are shown in Figure 2a,b. The array height is about 500 µm after an hourly synthesis (Figure 2a). The surface of the array consists of a dense network of entangled CNTs (Figure 2b). In some places, the CNT bundles protrude above the surface by several micrometers (marked with orange ovals in Figure 2a,b). A TEM study showed that the arrays contain MWCNTs with a fairly wide distribution over the outer diameter (Figure 2c). The average diameter is 40 nm, and the deviation from this value is 20 nm. The estimation was made using ten TEM images with hundreds of nanotubes. Dark rounded nanoparticles (shown by arrows in Figure 2c) correspond to the residual metal catalyst. These nanoparticles are located inside nanotubes at their closed tips; the inner channels of MWCNTs are mostly empty (Figure 2d). The nanotube walls consist of 10–14 coaxial graphitic-like layers.

The phase composition of MWCNTs was determined by powder XRD. The XRD pattern of MWCNTs vertically aligned to the substrate (curve 1 in Figure 3) is dominated by a peak at 2θ = 68°. This peak corresponds to the (100) silicon reflection. The peak at 2θ = 26.05° is formed due to X-ray diffraction from the graphitic layers of MWCNTs. The cylindrical structure of the layers causes peak asymmetry at small angles [55]. The interlayer distance determined from the position of the peak is ~3.42 Å. Iron nanoparticles present in the sample can contribute to the peaks between the (002) graphite reflection and the (100) silicon reflection. To identify these species present in MWCNTs, XRD analysis was performed on a sample without the substrate. Two arrays were carefully detached from the silicon substrates and applied on the surface of a background-free cuvette. A change in the direction of the X-ray beam relative to the MWCNTs leads to an increase in the intensity of the (002) graphite reflection (curve 2 in Figure 3). The peaks at 2θ = 37.75°, 42.95°, 43.80°, and 44.80° correspond to the iron carbide Fe_3_C phase as shown by comparison with the reference XRD pattern (PDF number card #72-1110, Figure 3). A higher relative intensity of the reflection at 42.95° in the pattern of the MWCNT sample can be related to the contribution of the (110) reflection from α-Fe. Weak reflections at 28.40° and 47.35° also correspond to this iron phase.

Raman spectrum of the MWCNT array showed bands D, G, and 2D (Figure 4a), characteristic of sp^2^-carbon materials. The absence of radial breathing modes appearing in the spectral region up to 500 cm^−1^ confirms that all CNTs are multilayered. The G band at 1580 cm^−1^ corresponds to tangential vibrations of carbon atoms that the honeycomb network is composed of. The D band is a sign of the presence of defects in this network. The ratio of intensities I_D_/I_G_ is commonly used to evaluate the density of defects in sp^2^-carbon. The I_D_/I_G_ value is 0.4 for the sample under study, and this indicates a high ordering of carbon atoms in the nanotube walls. A high relative intensity of the 2D band is also an indicator of the good structural quality of MWCNTs.

The atomic concentrations of elements on the surface of the MWCNT array were determined from the survey XPS spectrum measured at a photon energy of 830 eV (Figure 4b). In addition to carbon, the array contains oxygen (~1.5 at %). The spectrum shows no signal from iron; hence, its surface concentration is less than 1 at%. This is due to the covering of iron nanoparticles by several graphitic-like layers. The C 1s XPS spectrum exhibits an asymmetric peak at 284.5 eV from the sp^2^-carbon (Figure 4c). Weak components at 285.2 and 286.6 eV are assigned to defect states (non-hexagonal rings and vacancies) and oxidized carbon states, respectively. The NEXAFS C K-edge spectrum exhibits a π*-resonance at 285.3 eV and an σ*-resonance at 291.6 eV, which are characteristic of graphitic-like materials [56]. The XPS and NEXAFS data confirm that the synthesized arrays contain a negligible amount (if any) of amorphous carbon by-products and that the iron species are encapsulated.

### 3.3. Mechanism of CNT Array Growth

Firstly, we consider the initial stage of MWCNT growth. SEM images were taken in different registration modes (Figure 5a–c). 

SEM image registered using a reduced accelerating voltage of 2 kV shows that, as a result of synthesis for 120 s, the substrate surface is covered with a dense layer of nanoparticles and short nanotubes less than 500 nm long (Figure 5a). Metallic nanoparticles are identified in the image taken from the same place of the sample in the backscattering electron (BSE) mode (highlighted in red in Figure 5b). Iron is heavier than carbon and therefore scatters more electrons. Iron nanoparticles are visible on the substrate, and sometimes at the top of the nanotubes. All iron nanoparticles are encapsulated with carbon (highlighted in green). Iron nanoparticles located on the substrate provide the root growth of the MWCNTs. Nanoparticles observed at the top of the nanotubes may be responsible for the tip growth mechanism in the initial stage. When graphitic layers cover these nanoparticles, they do not participate in synthesis but contribute to the amount of iron present on the surface of the array. 

The average size of iron nanoparticles on the substrate surface was determined from the image obtained at an accelerating voltage of 5 kV and the registration of secondary electrons (Figure 5c). The increased accelerating voltage of the electrons makes it possible to see deeper layers under the carbon coating. Registration of secondary electrons provides a higher image resolution than when using a BSE detector. Image analysis shows that the average size of iron nanoparticles on the substrate is 20–30 nm, and the coating carbon thickness is about 10 nm.

The scheme of the initial stage of MWCNT growth on a substrate (Figure 5d) was proposed based on the analysis of SEM images. At elevated temperatures, ferrocene molecules decompose on the substrate surface, and iron atoms agglomerate into nanoparticles about 20 nm in size. Toluene molecules interact with these nanoparticles, decompose, and the nanoparticles become saturated with the released carbon. After that, the formation of MWCNTs with an outer diameter of 40 ± 20 nm begins. An iron nanoparticle can break in such a way that one part of it remains bound to the substrate, while the other part rises up due to the growing nanotube. Thus, in the initial stage of synthesis, the root and tip growth mechanisms of MWCNTs can be realized.

Further supply of ferrocene and toluene to the substrate leads to an increase in the size of catalyst nanoparticles and their continuous saturation with carbon. CNTs continue to grow, and excess iron is included in the nanotube cavities. This explains the linear increase in the array thickness in the second stage of synthesis (Figure 1b). It is possible that a certain amount of iron nanoparticles is formed on the walls and closed ends of CNTs; however, such nanoparticles are rapidly covered with carbon and lose their catalytic activity. With long-term synthesis, a discrepancy between the catalyst source feed rate and the CNT growth rate may occur. At an insufficient concentration of hydrocarbon vapors in the reaction zone, the growth of CNTs is suppressed or even stops. Even with a continuous supply of the reaction mixture, the oscillating growth of CNT arrays can be observed [33]. In this case, the CNT growth rate decreases and then increases again when the catalyst nanoparticles are saturated with carbon. Thus, to ensure the formation of arrays with long CNTs, it is necessary to match the regimes of partial pressures and temperatures, i.e., an accurate selection of synthesis parameters [57].

### 3.4. Distribution of Iron Nanoparticles in MWCNT Array

EDX spectroscopy was used to analyze the depth distribution of iron nanoparticles in MWCNT arrays. Since MWCNTs grow by the root mechanism, the upper part of the array contains structures synthesized in the first stage of synthesis. The middle part of the array is formed in the second stage of continuous synthesis at a constant nanotube growth rate (Figure 1d). The bottom part of the array (near the silicon substrate) corresponds to the final third stage of MWCNT growth.

For the SEM/EDX study, the MWCNT-coated silicon substrate was split to reveal the interior of the array. Figure 6a presents a SEM image of MWCNTs with several attached/encapsulated round nanoparticles present in the middle of the array. The image of the same part of the sample obtained in the combined mode, when secondary electrons and backscattered electrons were recorded, identifies many heavy components (highlighted in red) which should be attributed to metal nanoparticles (Figure 6b). Figure 6c shows the SEM image of the MWCNT array along its entire length and the distribution of iron nanoparticles in the top, middle, and bottom parts of the array. The images of different areas of the array were obtained in the combined registration mode at 15 kV. At a high accelerating voltage, secondary electrons are knocked out from a greater depth, which makes it possible to analyze the sample at a distance of several nanometers from the surface. In this case, metal atoms with a large mass become visible even under the surface carbon layer. SEM images of metal nanoparticles were analyzed using the ImageJ software, USA [58]. The average density of nanoparticles per square micrometer is 80 in the top part of the array, 100 in the middle of the array, and 120 in the part near the silicon substrate. In addition, EDX spectroscopy was used to determine the composition of various parts of the array. The data are given in Table 1 and confirm the lower iron concentration in the upper layer of the array and its increase in the bottom layer of the array.

EDX spectroscopy measurements were also carried out along a vertical line in the direction from the substrate to the sample surface. This line corresponded to an array thickness of 124 μm. The diameter of the scanning area was about 3 μm. The average depth of EDX analysis for a carbon material was about 3–5 µm at an accelerating voltage of 15 kV. Since the array material has a low bulk density of ~0.4 g/cm^3^, the probing depth can reach twenty micrometers. The change in the Fe/C ratio along the MWCNT array is shown in Figure 6d. The iron concentration gradually decreases with distance from the substrate surface. The data obtained indicate that the catalyst accumulates in the region near the substrate surface. This is due to the continuous supply of an iron source (in our case, ferrocene) to the reaction zone. 

## 4. Conclusions

The use of a video registration system to study the aerosol-assisted CCVD synthesis of MWCNTs vertically oriented on a substrate determined three main stages of their growth. In the first stage, the growth rate of MWCNTs is low due to the need to first form catalyst nanoparticles. In the process under study, they are formed as a result of ferrocene thermolysis. In the second stage, the MWCNT array grows linearly with time at a constant supply of toluene and ferrocene vapors. In the third stage of synthesis, the formed array limits the access of reagents to the substrate, where they decompose, and therefore, the growth of MWCNTs slows down. It was found that, at all stages of the synthesis, catalyst nanoparticles have a size of 20–30 nm. According to XRD data, the nanoparticles are iron carbide Fe_3_C with an admixture of α-Fe. The nanoparticles are encapsulated by several carbon layers, which prevents the observation of the iron signal in XPS. An EDX spectroscopy study revealed that the iron-to-carbon ratio gradually decreases from the bottom of the MWCNT array to its top. Our study showed that MWCNT arrays are not only anisotropic but also a gradient material, which can exhibit its unique structural properties when interacting with a magnetic field or electromagnetic microwave radiation. Since the concentration of iron-based nanoparticles depends on the synthesis conditions, it becomes possible to influence the distribution profile of iron particles, imparting the required functional properties to the material. The results obtained are of interest from the point of view of the possibility of using arrays of vertically aligned MWCNTs to create MEMS and magnetic memory elements. It was shown that MWCNT arrays with a thickness of millimeters and an area of 90 cm^2^ can be inexpensively produced using the aerosol-assisted CCVD method [59]. The industrial production of such arrays can be realized using a large reactor adopted for continuous synthesis.

## Figures and Tables

**Figure 1 materials-15-06639-f001:**
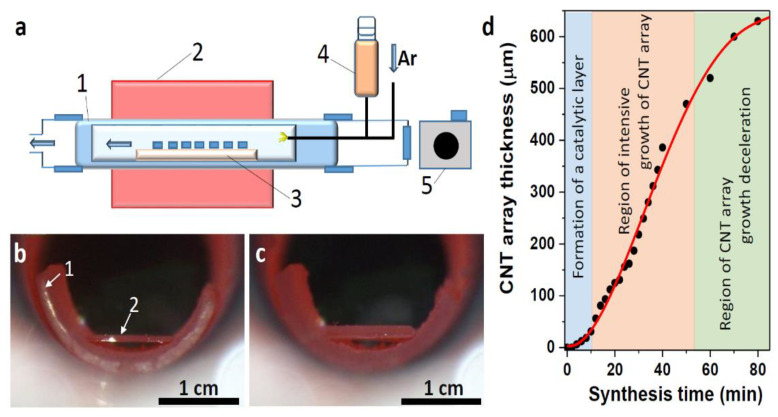
Scheme of the CVD reactor (**a**). The numbers on the scheme indicate: 1—quartz tube reactor, 2—high-temperature tubular furnace, 3—quartz boat with silicon substrates, 4—injection device, 5—video registration system. Image of the edge of a quartz boat with a silicon substrate after 4 min (**b**) and 40 min (**c**) of CNT synthesis. The numbers indicate: 1—quartz boat; 2—silicon substrate edge. The growth rate of the CNT array (**d**).

**Figure 2 materials-15-06639-f002:**
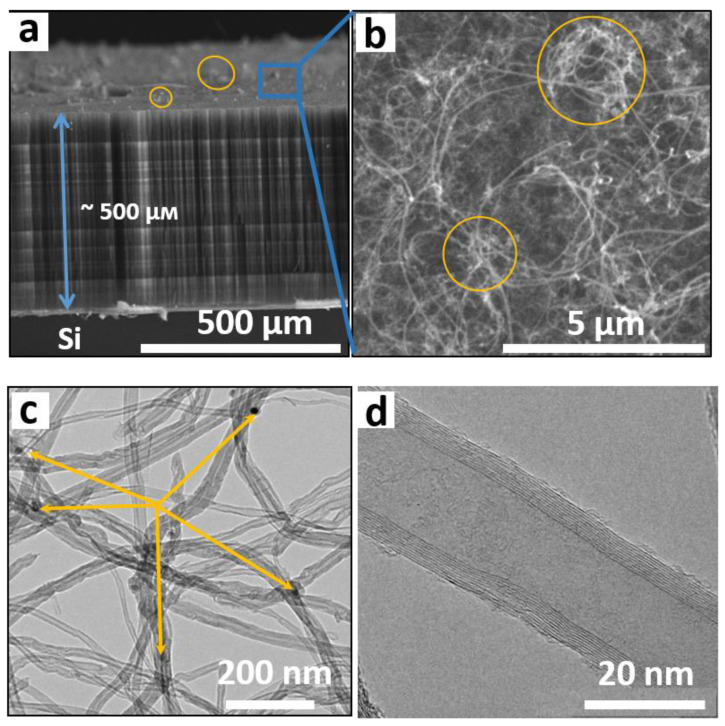
SEM images of side view (**a**) and top view (**b**) of CNT array. TEM images of CNTs taken with low magnification (**c**) and high magnification (**d**). Arrows in (**c**) show encapsulated iron nanoparticles.

**Figure 3 materials-15-06639-f003:**
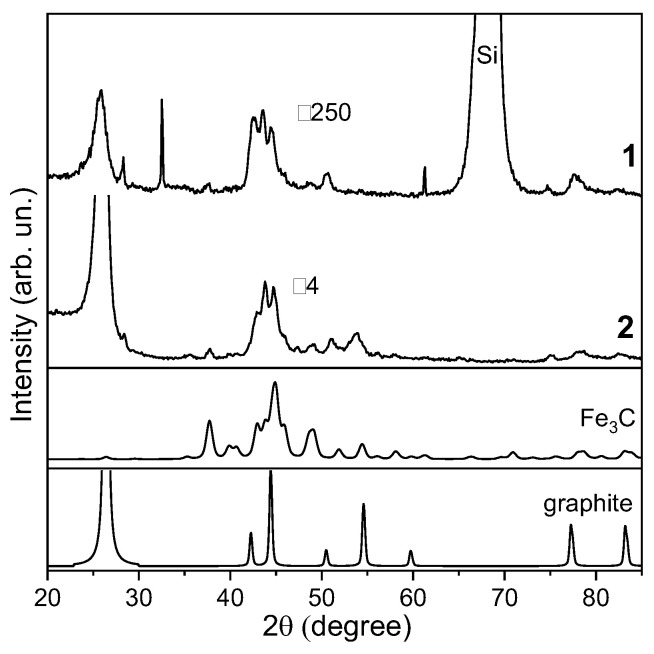
XRD patterns of MWCNT array on silicon substrate (1) and MWCNT sample detached from substrate (2) in comparison with reference patterns of iron carbide Fe_3_C and graphite. The patterns (1) and (2) are presented with zoom of 250 and 4, respectively.

**Figure 4 materials-15-06639-f004:**
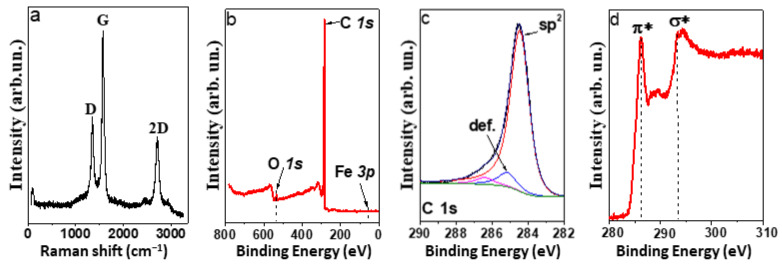
Raman spectrum (**a**), survey XPS spectrum (**b**), XPS C 1 s spectrum (**c**), and NEXAFS C K-edge spectrum (**d**) of MWCNT array.

**Figure 5 materials-15-06639-f005:**
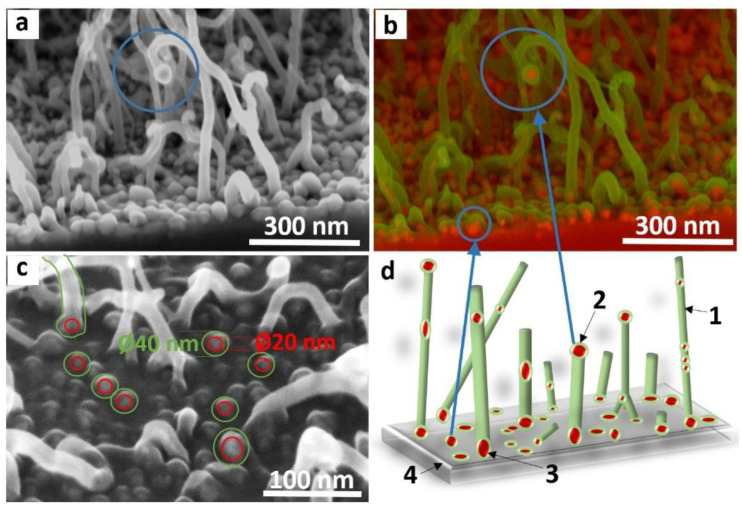
SEM images of catalytic layer and MWCNT array on silicon substrate obtained in the secondary electron registration mode at an accelerating voltage of 2 kV (**a**), in secondary (green color) and backscattering (red color) electron registration modes (**b**), and in secondary electron mode at 5 kV (**c**). Scheme of the initial growth stage of MWCNT array on silicon substrate (**d**). The numbers indicate: 1—MWCNT, 2—catalytic nanoparticle for tip growth; 3—catalytic nanoparticle for root growth; 4—silicon substrate.

**Figure 6 materials-15-06639-f006:**
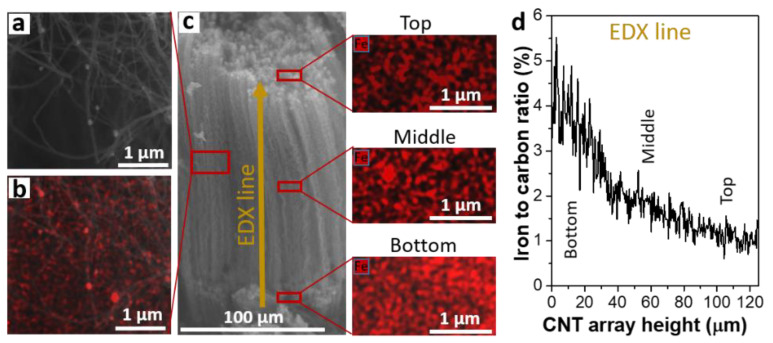
SEM image of MWCNTs obtained in secondary electron registration mode (**a**) and combined secondary electron/backscattering electron registration mode (**b**). SEM image of the array showing the areas of analysis and images of catalytic nanoparticles (red colored) (**c**). EDX-derived Fe/C ratio obtained using EDX analysis along the height of the MWCNT array (**d**).

**Table 1 materials-15-06639-t001:** Content (at.%) of main elements in various regions of the MWCNT array (Figure 6c) determined by EDX spectroscopy.

Regions	Carbon	Silicon	Iron	Oxygen
Top	90	2	4	4
Middle	88	5	5	2
Bottom	82	10	6	2

## Data Availability

Not applicable.

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
