# Peer review of "Distribution of Iron Nanoparticles in Arrays of Vertically Aligned Carbon Nanotubes Grown by Chemical Vapor Deposition"

_materials, 2022, doi:10.3390/ma15196639_

Round 1

Reviewer 1 Report

It is an interesting paper about the distribution of iron nanoparticles. The market mainly includes entangled single-walled CNTs (SWCNTs) or multi-walled CNTs (MWCNTs) produced by catalytic chemical vapor deposition (CCVD).

The study is original and it shows, that although the mixture of the precursors supplies evenly to the reactor, the iron content in the top of the array is lower and it increases with the approaching to the substrate.

At the same time, the technology of CCVD synthesis of arrays of 34 ordered CNTs is of great interest.

The authors are using two ways to prepare a catalyst for the synthesis of CNT arrays. The purpose of this work is to determine the main stages of aerosol CCVD synthesis of CNTs and to evaluate the distribution of metal nanoparticles in an array produced from a ferrocene-toluene mixture.

It is an interesting paper, but the references are old, some of them from 20 years ago, such 2001, 2002, 2004. My suggestion is to review and update the references. 

The structure of the paper is adequate, and I appreciate the well-written abstract with relevant information for the reader. The introduction presents the topic and aims, and gives an overview of the paper.

The recommendations and suggestions are:

  1. To develop the section of “Conclusions” with the interpretation of results, the implications for further research and challenges of Industry.
  2. To update the references, some of them are very old, such 20 years ago. There are more relevant papers for the research.
  3. In this paper the methods adequately described and the introduction provide sufficient background.

    The structure of the paper is adequate, and I appreciate the well-written abstract with relevant information for the reader. The introduction presents the topic and aims, and gives an overview of the paper, the text is clear and easy to read.

    The conclusions consistent with the evidence and arguments presented, but my suggestion is to develop the section of “Conclusions” with the interpretation of results, the implications for further research and challenges of Industry.

Author Response

The recommendations and suggestions are:

  1. To develop the section of “Conclusions” with the interpretation of results, the implications for further research and challenges of Industry.

Authors’ response: We have updated the Conclusions section. We added the sentences about the specificity of this work (the use of a video registration system to study the growth of nanotubes), the results obtained about the composition and structure of nanoparticles, the prospects of the use of Fe-filled MWCNT arrays and scaling of the synthesis. The sentences are highlighted in blue in the manuscript and are presented bellow:

“The use of a video-registration system to study of the aerosol-assistant CCVD synthesis of MWCNTs vertically oriented on a substrate determines three main stages of their growth.”

“According to the XRD analysis, the nanoparticles are iron carbide Fe3C with an admixture of α-Fe. The nanoparticles are encapsulated by several carbon layers, which prevents the observation of the iron signal in XPS. An EDX spectroscopy study reveals that the ratio of iron to carbon ratio increases four times from the bottom of the MWCNT array to the top.”

“The results obtained are of interest for potential applications of arrays of vertically aligned MWCNTs for the creation of MEMS and magnetic memory elements. It has been shown that the MWCNT arrays with thickness of millimeters and an area of 90 cm2 can be inexpensively produced using the aerosol-assisted CCVD method [49]. The industrial production of such arrays can be realized using a large reactor adopted for continuous synthesis.”

2. To update the references, some of them are very old, such 20 years ago. There are more relevant papers for the research.

Authors’ response: We modified the Introduction section by adding new sentences (highlighted in blue in the manuscript) and added recent papers in the field of the synthesis and applications of vertically aligned carbon nanotubes.

  1. Parmee, R.J.; Collins, C.M.; Milne, W.I.; Cole, M.T. X-ray generation using carbon nanotubes. Nano Converg. 2015, 2, 1–27. DOI: 10.1186/s40580-014-0034-2.
  2. Tovee, P.D.; Pumarol, M.E.; Rosamond, M.C.; Jones, R.; Petty, M.C.; Zezeb, D.A.; Kolosov O.V. Nаnoscale resolution scanning thermal microscopy using carbon nanotube tipped thermal probes. Phys. Chem. Chem. Phys. 2014, 16, 1174–1181. DOI: 10.1039/c3cp53047g.
  3. Bulusheva, L.G.; Arkhipov, V.E.; Fedorovskaya, E.O.; Zhang, S.; Kurenya, A.G.; Kanygin, M.A.; Asanov, I.P.; Tsygankova, A.R.; Chen, X.; Song, H.; Okotrub, A.V. Fabrication of free-stanfing aligned multiwalled carbon nanotube array for Li-ion batteries. J. Pow. Sour. 2016, 311, 42–48. DOI: 10.1557/JMR.2008.0356.
  4. Fedorovskaya, E.O.; Apartsin, E.K.; Novopashina, D.S.; Venyaminova, A.G.; Kurenya, A.G.; Bulusheva L.G.; Okotrub, A.V. RNA-modified carbon nanotube arrays regognizing RNA via electrochemical capacitance response. Mater. Design 2016, 100, 67–72. DOI: 10.1016/J.MATDES.2016.03.110.
  5. Xiao, D.; Wang, Q.; Wang, Z.; Zhang, Y.; Wu, J.; Fan, K.; Sun, L.; Zhu, M.; Ng, Z.K.; Teo, E.H.T.; Hu, F. Real-time THz beam profiling and monitoring via flexible vertically aligned carbo nanotube arrays. Adv. Optical Mater.2022, 2201363. DOI: 10.1002/adom.202201363.
  6. Huang, S.; Du, X.; Ma, M.; Xiong, L. Recent progress in the synthesis and applications of vertically aligned carbon nanotube materials. Nanotechnol. Rev. 2015, 10(1), 1592–1623. DOI: 10.1515/ntrev-2021-0102.
  7. Kohls, A.; Maurer Ditty, M.; Dehghandehnavi, F.; Zheng, S. Y. Vertically aligned carbon nanotubes as a unique material for biomedical applications. ACS Appl. Mater. Interfaces. 2022, 14(5), 6287–6306. DOI: 10.1021/acsami.1c20423.
  8. Cho, W.; Schulz, M.; Shanov, V. Growth termination mechanism of vertically aligned centimeter long carbon nanotube arrays. Carbon. 2014, 69, 609–620. DOI: 10.1016/j.carbon.2013.12.088.

   46. Das, R.; Bee Abd Hamid, S.; Eaqub Ali, M.; Ramakrishna, S.; Yongzhi, W.  Carbon nanotubes characterization by X-ray powder diffraction–a review. Curr. Nanosci. 2015, 11(1), 23–35. DOI: 10.2174/1573413710666140818210043.

  1. Meysami, S.S.; Koós, A.A.; Dillon, F.; Dutta, M.; Grobert, N. Aerosol-assisted chemical vapour deposition synthesis of multi-wall carbon nanotubes: III. Towards upscaling. Carbon 2015, 88, 148–156. DOI: 10.1016/j.carbon.2015.02.045.

3. In this paper the methods adequately described and the introduction provide sufficient background.

The structure of the paper is adequate, and I appreciate the well-written abstract with relevant information for the reader. The introduction presents the topic and aims, and gives an overview of the paper, the text is clear and easy to read.

The conclusions consistent with the evidence and arguments presented, but my suggestion is to develop the section of “Conclusions” with the interpretation of results, the implications for further research and challenges of Industry.

Authors’ response: We thank the Reviewer for the high evaluation of our work. A description of the modification of the Conclusion part is given in our answer to comment 1.

Reviewer 2 Report

Dear Editor:

The manuscript with {Distribution of iron nanoparticles in arrays of vertically aligned 2 carbon nanotubes grown by chemical vapor deposition} title explained the fabrication of carbon nanotubes and distribution of iron nanoparticles. The above manuscript is a valuable part of the work and could be accepted for publication in the " Materials" after these minor corrections:

1-The Abstract is too short; it is better to use more sentences.

2- The authors must use recent references.

3- In line 96 authors used toluene, why do authors use toluene?

4-Figures 2 a and b have to revise and the size of them is too small.

5- In the characterization of CNT arrays part, in my opinion, if the authors use IR and PXRD, it will be useful for the manuscript.

6- In the mechanism of the CNT array growth part authors should describe exactly about kind of morphology of the nanoparticles.

7-Authors are able to use ImageJ software to determine the size of nanoparticles in the SEM image.

8- EDX измерения распределения железа в массивах УНТ title is not clear.

9- English language of the manuscript needs to be revised by native people.

10- Authors could utilize such as some paper for a description of kind of morphology and type of application nanoparticles in result and discussion as well as introduction parts which are as follows: (a) Inorganica Chimica Acta 483 (2018): 516-526. (b)  Research on Chemical Intermediates 45.10 (2019): 5067-5089. 

Reviewer 3 Report

section 3.4: some different languages appeared.

Line 221, XXX microscope: the manuscript is not reviewed properly and submitted in haste.

colored SEM images are misleading. How can one see at the middle of the bundle from a side view without removing the top layer and similarly for the bottom layer?  

Apart from this, all other things has already been reported in several previous articles. 

Author Response

We thank the Reviewer for valuable comments and below we provide answers to them and describe the changes made to the manuscript.

section 3.4: some different languages appeared.

Authors’ response: We have corrected these moments.

Line 221, XXX microscope: the manuscript is not reviewed properly and submitted in haste.

Authors’ response: The types of all used microscopes are indicated in the current version of the manuscript in Materials and Methods.

Colored SEM images are misleading. How can one see at the middle of the bundle from a side view without removing the top layer and similarly for the bottom layer?  

Authors’ response: The use of an increased accelerating voltage of electrons in SEM microscopy makes it possible to knock out secondary electrons from the studied image from a greater depth. In this case, the surface layers (several nanometers) become, as it were, “transparent”. This point is clarified in the part 3.4. Distribution of iron nanoparticles in MWCNT array: “The images of different arears of the array were obtained in the combined registration mode at 15 kV. At a high accelerating voltage, secondary electrons are knocked out from a greater depth, which makes it possible to analyze the sample for several nanometers from the surface. In this case, metal atoms with a large mass become clear even under surface carbon layer.”

Apart from this, all other things has already been reported in several previous articles. 

Authors’ response: In previous works, the mechanism of growth of CNT arrays, synthesis parameters, features of the CVD reactor, etc. were described. However, a comprehensive study of the distribution of metal nanoparticles over an array of CNTs has not been carried out. Here we use a video registration system to determine the main stages of array growth and then analyze the array regions formed at these stages using the SEM/EDX method. This moment is highlighted in the first sentence of Conclusion: “The use of a video-registration system to study of the aerosol-assistant CCVD synthesis of MWCNTs vertically oriented on a substrate determines three main stages of their growth.” and further in this part: “An EDX spectroscopy study reveals that the ratio of iron to carbon ratio increases four times from the bottom of the MWCNT array to the top.”

Reviewer 4 Report

Dear author,

This study presents the mechanism of carbon nanotube growth via aerosol-assistant catalytic chemical vapor deposition.

Thre result is new and the scientific merit is broad.

However, the manuscript needs some amendments before publication.

  1. The reviewer needs to justify the originality of this study by describing soundness of the experimental procedure.
  2. The relationship with each data: The relationship of XPS data, SEM data and nanotube growth mechanism needs to be described.
  3. The reviewer recommends that the author describe the relationship of nanoparticle (size and spatial) distribution and carbon atoms, i.e. the characteristic of nanotube growth mechanism (relating to thermodynamics)
  4. The author needs to describe the relationship of movement of atoms (in Table 1) and nanotube growth mechanism. The insertion of schematic figure may improve the visuality.
  5. English quality: Some errors can be found in the presentation. 

Author Response

There result is new and the scientific merit is broad.

We are grateful to the Reviewer for the useful comments. The response is described below and the corresponding change in the manuscript is highlighted in blue.

However, the manuscript needs some amendments before publication.

1. The reviewer needs to justify the originality of this study by describing soundness of the experimental procedure.

Authors’ response: The originality of the experimental work lies in the use of the video registration system (shown in Figure 1a) during synthesis. This allowed to determine the main stages of array growth and analyze the array regions formed at these stages using the SEM/EDX method. This moment is highlighted in the first sentence of Conclusion: “The use of a video-registration system to study of the aerosol-assisted CCVD synthesis of MWCNTs vertically oriented on a substrate determines three main stages of their growth.” and further in this part: “An EDX spectroscopy study reveals that the ratio of iron to carbon ratio increases four times from the bottom of the MWCNT array to the top.”

2. The relationship with each data: The relationship of XPS data, SEM data and nanotube growth mechanism needs to be described.

Authors’ response: The purpose of this work is to study the distribution of iron nanoparticles in MWCNT arrays synthesized by aerosol-assisted CCVD method. It is indicated in the last paragraph of Introduction: “The purpose of this work is to determine the main stages of aerosol CCVD synthesis of CNTs and to evaluate the size and distribution of metal nanoparticles in an array produced from a ferrocene-toluene mixture.” This was done using the SEM/EDX method and the results obtained are useful for deeper understanding of the growth mechanism of MWCNTs. Using this data we described the mechanism as follows: “At elevated temperatures, ferrocene molecules decompose on the substrate surface, and iron atoms agglomerate into nanoparticles about 20 nm in size. Toluene molecules interact with these nanoparticles, decompose, and the nanoparticles are saturated with the released carbon. After that, the formation of MWCNTs with an outer diameter of 40±20 nm begins. An iron nanoparticle can break in such a way that one part of it remains bound to the substrate, while the other part rises up due to the growing nanotube. Thus, at the initial stage of synthesis root and tip growth mechanisms of MWCNTs can be realized.

Further supply of iron and toluene to the substrate leads to an increase in the size of catalyst nanoparticles and their continuous saturation with carbon. CNTs continue to grow and excess iron is included in the nanotube cavities. This explain the linear increase in the array thickness at the second stage of synthesis (Figure 1b). We cannot exclude that a certain amount of iron nanoparticles is formed on the walls and tip ends of CNTs; however, such nanoparticles are rapidly covered with carbon and loss their catalytic activity. With long-term synthesis, a discrepancy between the catalyst source feed rate and the CNT growth rate may occur. With an insufficient concentration of hydrocarbon vapors in the reaction zone, the growth of CNTs is suppressed or even stops. Even with a continuous supply of the reaction mixture, the oscillating growth of CNT arrays can be observed [30]. In this case, the CNT growth rate decreases and then increases again when the catalyst nanoparticles are saturated with carbon.”

3. The reviewer recommends that the author describe the relationship of nanoparticle (size and spatial) distribution and carbon atoms, i.e. the characteristic of nanotube growth mechanism (relating to thermodynamics)

Authors’ response: In this experimental work, we note the fact of a gradient distribution of iron nanoparticles in MWCNT arrays. It is quite possible that the thermodynamic model can be used to describe non-uniform distribution of nanoparticles along the array height. However, this is not the aim of this study and can be done in future works.

4. The author needs to describe the relationship of movement of atoms (in Table 1) and nanotube growth mechanism. The insertion of schematic figure may improve the visuality.

Authors’ response: The initial stage of the MWCNT growth is illustrated in Figure 5c. A description of the proposed mechanism for the vertical growth of nanotubes on a substrate is provided in answer to comment 2.

5. English quality: Some errors can be found in the presentation. 

Authors’ response: The typos have been corrected.

Reviewer 5 Report

The present study shows the  distribution of iron nanoparticles in arrays of vertically aligned 2 carbon nanotubes grown by chemical vapor deposition . I would like to suggest few comments before its acceptance, as follows.

1. If possible, author can add XRD spectrum of CNT array with appropriate discussion (Fig.3).

2. Fig. 1d, 3d, & 5d should be improved because it is not visible to see/review.

3. Author claimed "The CNT arrays with a specific distribution of metal nanoparticles inside can be useful for various applications where magnetic properties are decisive", I can't see any related magnetic studies (for instance, VSM) of their sample. 

Author Response

The present study shows the  distribution of iron nanoparticles in arrays of vertically aligned 2 carbon nanotubes grown by chemical vapor deposition . I would like to suggest few comments before its acceptance, as follows.

We are grateful to the Reviewer for the comments and below provide answers.

  1. If possible, author can add XRD spectrum of CNT array with appropriate discussion (Fig.3).

Authors’ response: The XRD data was added as Figure 3 and described as: “The phase composition of MWCNTs was determined by powder XRD. The XRD pattern of MWCNTs vertically aligned to the substrate (curve 1 in Figure 3) is dominated by a peak at 2θ = 68°. This peak corresponds to the (100) silicon reflection. The peak at 2θ = 26.05° is formed due to X-ray diffraction from the graphitic layers of MWCNTs. The cylindrical structure of the layers causes peak asymmetry at small angles [46]. The interlayer distance determined from the position of the peak is ~3.42 Å. Iron nanoparticles present in the sample can contribute to the peaks between the (002) graphite reflection and the (100) silicon reflection. To identify these species present in MWCNTs, XRD analysis was performed on a sample without substrate. Two arrays were carefully detached from the silicon substrates and applied on the surface of a background-free cuvette. A change in the direction of the X-ray beam relative to the MWCNTs leads to an increase of the intensity of the (002) graphite reflection (curve 2 in Figure 3). The peaks at 2θ = 37.75°, 42.95°, 43.80°, 44.80° corresponds to the iron carbide Fe3C phase as shown by comparison with the reference XRD pattern (PDF number card #72-1110, Figure 3). A higher relative intensity of the reflection at 42.95° in the pattern of the MWCNT sample can be related with the contribution of the (110) reflection from α-Fe. Weak reflections at 28.40° and 47.35° are also correspond to this iron phase.”

  1. Fig. 1d, 3d, & 5d should be improved because it is not visible to see/review.

Authors’ response: The abovementioned figures were improved.

  1. Author claimed "The CNT arrays with a specific distribution of metal nanoparticles inside can be useful for various applications where magnetic properties are decisive", I can't see any related magnetic studies (for instance, VSM) of their sample. 

Authors’ response: Such studies are planned in future that is indicated at the end of Conclusions: “The results obtained are of interest for potential applications of arrays of vertically aligned MWCNTs for the creation of MEMS and magnetic memory elements.”

We also mentioned magnetic studies on similar samples in Introduction: “Being in a liquid state during synthesis, the metal penetrates into the internal cavity of CNTs and after cooling the sample forms nanoparticles. These nanoparticles are mainly in the form of Fe3C and α-Fe [30] and impart magnetic properties to arrays [31–34].”

Round 2

Reviewer 3 Report

The detection limit of EDS is not good for low atomic number elements, so the conclusive sentence  (about the ration of Fe and C) should be appropriately written. 

Reviewer 5 Report

Author responses to reviewer comments are satisfactory and the revised manuscript can be accepted for publication. 
